# Development and Validation of a Novel Robot-Based Assessment of Upper Limb Sensory Processing in Chronic Stroke

**DOI:** 10.3390/brainsci12081005

**Published:** 2022-07-29

**Authors:** Leen Saenen, Jean-Jacques Orban de Xivry, Geert Verheyden

**Affiliations:** 1Department of Rehabilitation Sciences, KU Leuven, 3001 Leuven, Belgium; geert.verheyden@kuleuven.be; 2Department of Movement Sciences, KU Leuven, 3001 Leuven, Belgium; jj.orban@kuleuven.be; 3KU Leuven Brain Institute, KU Leuven, 3000 Leuven, Belgium

**Keywords:** stroke, upper limb, somatosensation, sensory processing, assessment, robotics

## Abstract

Upper limb sensory processing deficits are common in the chronic phase after stroke and are associated with decreased functional performance. Yet, current clinical assessments show suboptimal psychometric properties. Our aim was to develop and validate a novel robot-based assessment of sensory processing. We assessed 60 healthy participants and 20 participants with chronic stroke using existing clinical and robot-based assessments of sensorimotor function. In addition, sensory processing was evaluated with a new evaluation protocol, using a bimanual planar robot, through passive or active exploration, reproduction and identification of 15 geometrical shapes. The discriminative validity of this novel assessment was evaluated by comparing the performance between healthy participants and participants with stroke, and the convergent validity was evaluated by calculating the correlation coefficients with existing assessments for people with stroke. The results showed that participants with stroke showed a significantly worse sensory processing ability than healthy participants (passive condition: *p* = 0.028, Hedges’ g = 0.58; active condition: *p* = 0.012, Hedges’ g = 0.73), as shown by the less accurate reproduction and identification of shapes. The novel assessment showed moderate to high correlations with the tactile discrimination test: a sensitive clinical assessment of sensory processing (r = 0.52–0.71). We conclude that the novel robot-based sensory processing assessment shows good discriminant and convergent validity for use in participants with chronic stroke.

## 1. Introduction

Upper limb somatosensory impairments are common after stroke and associated with decreased functional performance [1]. Somatosensory function is generally divided into three modalities, namely exteroception, proprioception and sensory processing [2]. Exteroception and proprioception are defined as the primary perceptual functions, while sensory processing is the secondary function requiring higher cortical processing of the primary modalities to interpret and discriminate between stimuli [2]. Unlike exteroception, which shows nearly full recovery after stroke [3], proprioception and sensory processing often remain impaired in the chronic phase after stroke. Proprioception is still impaired in up to 50% of participants at six months after stroke, while sensory processing is still impaired in about 22–28%, depending on the assessment used [4,5].

A systematic review from 2014 showed that somatosensory impairments are associated with motor function and functional performance [6], for example, with sensory processing being the second strongest predictor for functional outcome at 6 months after stroke, only preceded by muscle strength [7]. More recent studies [4,8,9,10,11,12,13] have confirmed these findings. Low to moderate correlations were found between somatosensory impairment and functional outcome [8,12]. Interestingly, sensory processing was found to be a prognostic factor for bimanual performance in mildly affected participants with chronic stroke [13]. Others have also suggested that recovery of somatosensory impairment might be a prerequisite for full motor recovery of the upper limb [11].

Given the persistence of somatosensory impairments into the chronic stage after stroke, and their importance for motor and functional recovery, it is key to accurately assess these impairments. Current clinical scales have suboptimal psychometric properties, such as having coarse ordinal scoring and ceiling effects [14]. For this reason, robot-based assessments have been recommended to assess upper limb impairments after stroke [15]. Various robot-based assessments of proprioception have been validated for use in participants with stroke [16,17,18], but we are not aware of any robot-based assessment for sensory processing.

The aim of this study was to develop a robot-based assessment of upper limb sensory processing and to provide an easily interpretable outcome by performing a factor analysis on the different robot parameters. We hypothesized that all robot parameters would be related to the same latent factor, indicating the overall sensory processing ability. In addition, we aimed to assess the discriminative validity and convergent validity of the novel assessment. To establish the discriminative validity, the novel test should find worse performance in participants with stroke compared to healthy participants. To establish convergent validity, the novel test should show high correlations with other assessments of sensory processing, while lower correlations should be found with assessments of exteroception, proprioception, motor function and performance. We hypothesized to find good discriminative and convergent validity of the novel assessment.

## 2. Materials and Methods

### 2.1. Participants

A flowchart of participant inclusion can be found in Figure 1. Sixty healthy participants and twenty participants with chronic stroke participated in this cross-sectional study (see Table 1 for participant characteristics). Healthy participants were included if they were aged 18 years and above, had no history of stroke or transient ischemic attack and did not present with upper limb sensorimotor impairments. Participants with stroke were included if they were 18 years or older, at least six months after a first-ever unilateral supratentorial stroke (as defined by the World Health Organization) and able to perform at least some shoulder abduction and wrist extension against gravity. They were excluded if they presented with any other neurological or musculoskeletal disorders, or severe communication and cognitive deficits. This study was registered at clinicaltrials.gov (NCT04721561). Since the results obtained with this exploratory study will inform future power-based studies, a sample size of 60 healthy participants and 20 participants with chronic stroke was deemed sufficient.

### 2.2. Experimental Set-Up

For the robot-based assessments, the Kinarm End-Point Lab (BKIN Technologies Ltd., Kingston, ON, Canada) was used. This bimanual end-point robot allows 2D movement in the horizontal plane, without anti-gravity support, while permitting control of visual feedback through a virtual reality screen. The robot collects positional data of both upper limbs at a rate of 1 kHz. All tests were performed in a seated position with bilateral trunk restraints to avoid compensatory trunk movements. A black cloth prevented vision of the upper limbs. In three participants, hand fixation was used to maintain hand position of the affected limb, due to limited grasp function.

### 2.3. Experimental Task

Sensory processing was evaluated using two versions of a three-step sensory processing task, which differed only in their first step. For the passive condition of the task (Figure 2A), the robot first passively moved the participant’s affected arm (or nondominant arm for healthy participants) in the shape of a triangle, tetragon or pentagon by starting from and returning to a starting point positioned 20 cm in front of the shoulder. For the active condition (Figure 2B), the participant was asked to explore the same shapes by moving the affected or nondominant arm between virtual walls delimiting the shape, which are described below. For both conditions, there was no visual feedback of shape or hand position. The participant was then asked to reproduce the shape without mirroring with the less affected or dominant arm within 15 s, by starting from and returning to the same starting position. Here, visual feedback was provided on the hand position but not on the reproduced path. Finally, the participant was asked to identify the explored shape out of six options presented on the screen of the Kinarm robot. Both conditions consisted of 15 randomized trials and were preceded by 5 practice trials (all shapes are provided in the Appendix A). Feedback on task performance was only provided during the practice trials. In the passive condition, the robot used a bell-shaped speed profile with a maximum speed of 0.67 m/s. In the active condition, the shape was delimited with use of position-dependent force regions. Along the lines of the shape, a zero-force region with a width of 0.2 cm existed in which the participant could actively move. Outside these lines, a virtual wall with a stiffness of 6000 N/m and a viscosity of −50 Ns/m was applied. Participants were allowed to explore each shape once, at a self-determined speed within a time limit of 30 s. For both conditions, each line of the explored shapes was between 2.92 cm and 14.14 cm in length.

### 2.4. Other Robot-Based Assessments

To assess motor function, a 4-target visually guided reaching test was performed with each arm separately [19]. In this test, participants were required to perform center-out reaching movements as quickly and as accurately as possible. Ten outcome parameters were calculated, covering reaction time, speed and accuracy of reaching, after which they were combined into a single task score with higher values meaning worse motor function [19,20]. To assess proprioception, a 9-target arm position matching test was performed [16]. In this test, the robot passively moved the participant’s affected or non-dominant arm, after which the participant was asked to actively mirror this position with their other arm. Twelve outcome parameters were calculated, including magnitude and variability of position errors, and combined into a single task score with higher values meaning worse proprioception [16,20]. In the visually guided reaching test, visual feedback of the hand position was provided, while in the arm position matching test, visual feedback was completely blocked. No practice trials were performed, but good understanding of instructions was checked by the examiner. Both tests showed good reliability and validity in participants with stroke [16,19,21].

### 2.5. Clinical Assessments

Clinical assessments were performed on function and activity levels of the International Classification of Functioning, Disability and Health [22]. An overview of all clinical assessments can be found in Table 2.

### 2.6. Data Analysis

For the robot-based sensory processing task, position and velocity data of both upper limbs were imported into MATLAB (MathWorks, Natick, MA, USA). The start of the exploration and reproduction phases was selected based on a hand velocity threshold of 0.02 m/s after leaving the starting point. This threshold was established based on a close examination of pilot data and aimed to exclude postural oscillations of the hand while at the starting point. Both phases ended when the hand reached the starting point again. Hand position data were normalized in time (‘interp1’ in MATLAB) to ignore speed differences between the exploration and reproduction phase. To evaluate reproduction accuracy, three parameters were calculated using custom MATLAB scripts. First, we computed cross-correlation values (‘xcorr’ in MATLAB) between the horizontal or vertical normalized hand position signals from the explored and reproduced shapes. Cross-correlation values ranged between −1 and 1, with higher values indicating larger similarity. Next, we calculated the dynamic time warping parameter (‘dtw’ in MATLAB) between the explored and reproduced shapes, by representing both shapes as two temporal sequences of X and Y hand position signals and finding optimal alignment between them irrespective of speed. Dynamic time warping values equalled the distance between the two sequences, with higher values indicating less similarity. A Procrustes analysis (‘procrustes’ in MATLAB) compared the similarity between explored and reproduced shapes by optimally superimposing both shapes by translating, rotating and scaling the reproduced shape on top of the explored shape. Procrustes values indicated the distance between both superimposed shapes, with higher values indicating less similarity. Finally, we calculated the percentage of correctly identified shapes during the identification phase. Certainty of the participant’s answer during the identification phase was evaluated using a 4-point Likert scale ranging from 0 to 3, with higher values indicating higher certainty.

### 2.7. Statistical Analysis

All statistical analyses were performed in R version 4.0.3 (R Foundation, Vienna, Austria) [41]. Statistical tests were performed two-tailed with an alpha level of 0.05. 

Because the Shapiro–Wilk test indicated a non-normal distribution for most outcomes, we compared participant characteristics between healthy participants and participants with stroke using Mann–Whitney U tests and Fisher’s exact tests (‘shapiro.test’, ‘wilcox.test’, and ‘fisher.test’ from the stats package [41], respectively). 

To combine all five parameters of the robot-based sensory processing assessment into one factor score, an exploratory factor analysis using the principal factor method [42] was performed for the passive and active conditions separately. This analysis was performed on the data of healthy participants using the ‘fa’ function from the psych package [43]. Scree plots indicated one latent factor, which was defined as the sensory processing ability. The factor scores of healthy participants were calculated using the regression method for the passive and active conditions separately. We then obtained factor scores for participants with stroke by first calculating the standard scores of all five parameters against the mean and standard deviation of healthy participants, and then calculating the weighted mean of these standard scores by using the factor loadings of the exploratory factor analysis as weights. This way, a factor score of zero equals the mean performance of healthy participants, and the scores of participants after stroke can be interpreted as the magnitude of deviation from normal performance.

To evaluate the discriminative validity, we compared the performance between healthy participants and participants with stroke. A robust three-way ANOVA based on 20% trimmed means (‘bwwtrim’ from Wilcox 2017 [44]) was performed on cross-correlation values using the participant group (healthy participants vs. participants with stroke) as a between-group factor, and task condition (active vs. passive) and axis direction (X vs. Y) as the within-group factors. A robust two-way ANOVA based on 20% trimmed means (‘bwtrim’ from Wilcox 2017 [44]) was performed on dynamic time warping parameters, on the outcomes of the Procrustes analysis and on the percentage of identified shapes, with the participant group as the between-group factor and task condition as the within-group factor. When no interaction effect was present, we reported the main effects. When an interaction effect was significant, the simple main effects were evaluated. We corrected for multiple comparisons using the Holm–Bonferroni method whenever simple main effects were calculated (‘p.adjust’ from the stats package [41]) [45]. For all ANOVA analyses, we reported the effect sizes as generalized eta squared [46,47] (‘anova_summary’ from the rstatix package [48]). We compared the factor scores of the passive and active conditions between healthy participants and participants with stroke with independent t-tests (‘t.test’ from the stats package [41]). Normality and homoscedasticity were confirmed a priori using Shapiro–Wilk tests and F-tests of equality of variances (‘shapiro.test’ and ‘var.test’ from the stats package [41]). We calculated effect sizes using Hedges’ g with the ‘cohen.d’ function from the effsize package [47,49]. In addition, we compared the performance of all participants with stroke with 95% confidence intervals of healthy participants, to identify participants presenting with abnormal sensory processing ability, as was done previously by others [16,17,19].

The convergent validity was evaluated by calculating 20% Winsorized correlation coefficients (‘wincor’ from Wilcox 2017 [44]) for participants with stroke between outcomes on the robot-based sensory processing task, and standardized clinical and robot-based assessments of somatosensory function, motor function, cognitive function and activities. The strength of correlation was interpreted as follows: r_W_ < 0.30 = negligible correlation; r_W_ = 0.30–0.50 = low correlation; r_W_ = 0.50–0.70 = moderate correlation; r_W_ > 0.70 = high correlation [50]. In addition, we calculated 95% confidence intervals for all correlation coefficients by performing a Fisher z’ transformation (‘CIr’ from the psychometric package [51]) [52]. 

## 3. Results

Sixty healthy participants and twenty participants with chronic stroke were evaluated for their sensory processing abilities, by means of robot-based passive or active exploration, reproduction and identification of different shapes. The mean time needed to perform the passive and active conditions was 6.91 and 8.59 min, respectively.

### 3.1. Participants with Stroke Were Less Accurate in Reproducing the Explored Shapes

The cross-correlation values were worse in participants with stroke (mean 0.82 (SD 0.07)) than in healthy participants (mean 0.85 (SD 0.05); Figure 3A; main effect of group: F (1,72) = 3.83, *p* = 0.051, η^2^_G_= 0.06). We also found worse cross-correlation values in the active condition than in the passive condition (Figure 3A; main effect of condition: F (1,72) = 100.65, *p* < 0.001, η^2^_G_ = 0.21). We found no evidence that cross-correlation values were different on the X and Y axes (Figure 3A; main effect of direction: F (1,72) = 0.39, *p* = 0.534, η^2^_G_ < 0.01). There was neither a significant three-way interaction between participant group, task condition and axis direction (Figure 3A; F (1,72) = 3.22, *p* = 0.073, η^2^_G_ < 0.01), nor any two-way interactions (Figure 3A; group x condition: F (1,72) = 0.24, *p* = 0.622, η^2^_G_ < 0.01; group x direction: F (1,72) = 1.24, *p* = 0.266, η^2^_G_ < 0.01; condition x direction: F (1,72) = 3.14, *p* = 0.077, η^2^_G_ < 0.01).

Unlike cross-correlation, which looks at each axis direction separately, dynamic time warping and Procrustes analysis values encompass the shape as one entity. Dynamic time warping showed significantly worse values for participants with stroke compared to healthy participants (Figure 3B; main effect of group: F (1,76) = 5.02, *p* = 0.029, η^2^_G_ = 0.03). Participants with stroke (mean 185.96 (SD 79.31)) did not reproduce the shape as well as healthy participants (mean 155.58 (SD 78.98)). Dynamic time warping values were significantly worse for the active condition compared to the passive condition (Figure 3B; main effect of condition: F (1,76) = 79.11, *p* < 0.001, η^2^_G_ = 0.17), but there was no evidence that the group difference was influenced by task condition (Figure 3B; group x condition: F (1,76) = 0.42, *p* = 0.518, η^2^_G_ < 0.01). For the Procrustes analysis, the group difference was influenced by task condition (Figure 3C; group x condition: F (1,76) = 4.88, *p* = 0.031, η^2^_G_ = 0.02). In both conditions, participants with stroke showed slightly worse values than healthy participants, and the largest difference was found for the active condition. However, for both conditions, the difference was not significant (Figure 3C; passive condition: F (1,76) = 0.60, *p* = 0.554; active condition F (1,76) = 1.87, *p* = 0.166). In addition, the active condition showed significantly worse values than the passive condition for both participant groups (Figure 3C; healthy participants: F (1,76) = 7.08, *p* < 0.001; participants with stroke: F (1,76) = 6.77, *p* < 0.001), and the difference was largest in participants with stroke. Results from the Procrustes analysis also showed that reproduced shapes were larger than explored shapes, with a mean scale of 1.41 for both participant groups. Hand speed during the exploration of shapes did not differ between participant groups or task conditions and averaged 0.04 m/s (SD 0.01).

### 3.2. Participants with Stroke Identified Less Shapes than Healthy Participants, despite Being Equally Certain about Their Answers

After shape reproduction, we asked participants to identify the explored shape among six options presented on the screen. During this identification phase, participants with stroke (mean 43.66% (SD 20.17)) identified significantly less shapes compared to healthy participants (mean 62.92% (SD 19.94); Figure 3D; main effect of group: F (1,76) = 13.36, *p* < 0.001, η^2^_G_ = 0.15). The data did not provide any evidence of a difference between conditions (Figure 3D; main effect of condition: F (1,76) = 2.53, *p* = 0.118, η^2^_G_ = 0.02), or that the group difference was influenced by the task condition (Figure 3D; participant x condition: F (1,76) = 0.43, *p* = 0.499, η^2^_G_ < 0.01). Participants from both groups showed moderate certainty about their answers, with a mean certainty score of 2.16 (SD 0.48) in healthy participants and 1.99 (SD 0.53) in participants with stroke (main effect of group: F (1,76) = 1.63, *p* = 0.207, η^2^_G_ = 0.02). There was no evidence that condition influenced group differences (group x condition: F (1,76) = 0.11, *p* = 0.737, η^2^_G_ < 0.01).

### 3.3. Participants with Stroke Showed Worse Sensory Processing Ability 

We performed an exploratory factor analysis on all parameters to generate an easily interpretable outcome. This analysis indicated one latent factor, representing the sensory processing ability, and expressed as the factor score. All five parameters contributed to the factor score, and their factor loadings are shown in Table 3.

The factor score was significantly worse in participants with stroke than in healthy participants, and this was true for both conditions (Figure 3E; passive condition: t (78) = 2.25, *p* = 0.028, Hedges’ g = 0.58; active condition: t (78) = 2.83, *p* = 0.012, Hedges’ g = 0.73).

### 3.4. Identification of Abnormal Performance in Participants with Stroke

Based on the factor score, 11 participants with stroke (55%) had an impaired sensory processing ability on both the passive and active condition of the sensory processing task (Table 4). The percentage of correctly identified shapes showed the largest group of participants with abnormal performance, namely 16 and 15 participants (80% and 75%) for the passive and active condition, respectively (Table 4).

### 3.5. The Robot-Based Sensory Processing Task Was Moderately to Highly Correlated with Sensitive Clinical Tests of Sensory Processing

Convergent validity was established by correlating the factor scores of the robot-based sensory processing task with clinical and robot-based assessments of somatosensory function, motor function, cognitive function and activities. We found moderate to high correlations with sensitive clinical tests of sensory processing, in contrast to low correlations with tests of exteroception and proprioception (Table 5). In addition, low to moderate correlations were found with motor function and performance (Table 5). Correlation coefficients of all parameters with clinical and robot-based assessments can be found in the Appendix A, as well as all scatterplots (Appendix A).

## 4. Discussion

In this study, we developed and validated a novel robot-based sensory processing assessment based on passive or active exploration, reproduction and identification of different shapes. First, the discriminative validity was established by showing a significantly worse sensory processing ability in participants with chronic stroke compared to healthy participants, as revealed by the less accurate reproduction and identification of explored shapes. Second, the convergent validity was established by showing moderate to high correlations with sensitive clinical tests of sensory processing, low correlations with clinical and robot-based tests of exteroception and proprioception and low to moderate correlations with motor function and performance.

These novel robot-based assessments show some clear advantages compared to standard clinical assessments. They involve objective evaluation using sensitive outcome parameters measured on a continuous scale; therefore, no ceiling effects are present. Furthermore, a factor analysis creates the potential to simplify complex outcome parameters by calculating overlapping factor scores, in order to provide subsequent analyses which are easier to interpret.

Regarding convergent validity, it is important to keep the differences between robot-based and clinical assessments in mind. We found low correlations with the sharp-blunt discrimination subscale of the Erasmus modified Nottingham sensory assessment, the stereognosis section of the original Nottingham sensory assessment and the functional tactile object recognition test. However, these clinical assessments showed a clear ceiling effect and ordinal scaling, whereas the robot-based factor score did not (see Appendix A). Higher correlation coefficients were found with the tactile discrimination test, which is a more sensitive test without ceiling effects. Furthermore, the robot-based sensory processing assessment showed smaller associations with exteroception and proprioception than expected, even though the task requires processing of these primary functions. However, the reported correlations are in line with results found by Connell and colleagues, who found low to moderate agreement (kappa = −0.1–0.54) between the different modalities [53]. These results may indicate that, even though sensory processing uses the primary exteroceptive and proprioceptive information, it should be viewed as a distinct modality. Finally, we found low to moderate correlations with motor function and performance, suggesting the association between sensory processing and functional abilities after stroke, which has been reported by others before [8,12]. Correlations with clinical assessments of proprioception and motor function were similar to the correlations with robot-based assessments of these functions, indicating the robustness of the results.

Recently, Ballardini and colleagues developed a technology-based evaluation of sensory processing in a limited group of healthy participants and participants with chronic stroke [54]. The described protocol evaluates sensory processing based on exteroceptive information [54], while our protocol is based on the processing of mainly proprioceptive information. A similar task to ours was used in the experiment of Henriques and colleagues in 2004, where six healthy participants were asked to actively explore and reproduce tetragons using a planar robot [55]. Here, healthy participants were relatively accurate in reproducing the explored shapes; however, they consistently overestimated the size (mean scale of 1.15) [55], which is similar to the results found in this study. To the best of our knowledge, such a robot-based approach has never been used or validated in a group of participants with stroke. Therefore, the results from the present study add a novel assessment paradigm to the field of upper limb sensory processing evaluation.

The novel robot-based assessments do have some limitations. First, the reproduction phase requires contralateral arm movement; hence, the interhemispheric transfer of information is required, which might be disturbed after stroke. Second, reproduction is performed with the ipsilesional upper limb, which might show subtle but significant impairments [56]. Third, because of the non-simultaneous execution of the exploration, reproduction and identification phases, information has to be stored in the working memory, which is often impaired after stroke [57]. Still, our novel evaluation paradigm showed valid results in our subgroup with stroke, suggesting applicability in further research. Finally, because of a possible increase in performance accuracy through learning, an additional analysis was performed to evaluate for the learning effects of the novel sensory processing task, which can be found in the Appendix A.

It is important to acknowledge the limitations of this study. First, only a limited group of participants with chronic stroke was included, which reduces the generalizability of the results. As a result of this sample size, there was low power to compare the results between subgroups of participants with stroke with and without clinically diagnosed sensory processing deficits. However, an additional subgroup analysis was performed which can be found in the Appendix A. This additional analysis showed some interesting results and can guide further research. There also remains a lot of uncertainty about the magnitude of the correlation coefficients, given the large confidence intervals. Future research should therefore replicate results in a larger sample. As a second limitation, due to the set-up of the Kinarm robot and the active condition, which required active grasping of the end-point handles and active shoulder and elbow movements, only mild to moderately affected participants with stroke were eligible to participate. Indeed, in the present study, we found mostly only mild upper limb impairments. This might have led to an underestimation of the severity of sensory processing deficits in the general stroke population on the one hand, but this also limits the generalizability of results to the general stroke population, on the other hand. Future studies should therefore assess the usefulness and feasibility of the proposed measures in a more severely affected population. A robotic set-up which allows anti-gravity support might be preferred over the current end-point set-up for use in the more severely affected population.

Based on the results described here, suggestions can be made for further implementation of the novel robot-based sensory processing assessments. Despite its requirement of active movement of the affected arm, and therefore restricted use to mild to moderately impaired participants, the active condition might be preferred over the passive condition given its greater discriminative and convergent validity (i.e., greater difference between healthy participants and participants with stroke (Figure 3), and larger correlation coefficients with clinical tests of sensory processing (Table 5)). In addition, when the primary aim is to identify upper limb sensory processing deficits, it can be suggested to skip the reproduction phase, as the identification parameters showed more favorable discriminative validity results compared to the reproduction parameters (Figure 3 and Table 4). However, future research should prioritize replication of the current results in a larger and more heterogenous sample, and should include an additional evaluation of the reliability and responsiveness of both the robot-based passive and active sensory processing assessments, as advised by the COSMIN initiative [58].

## Figures and Tables

**Figure 1 brainsci-12-01005-f001:**
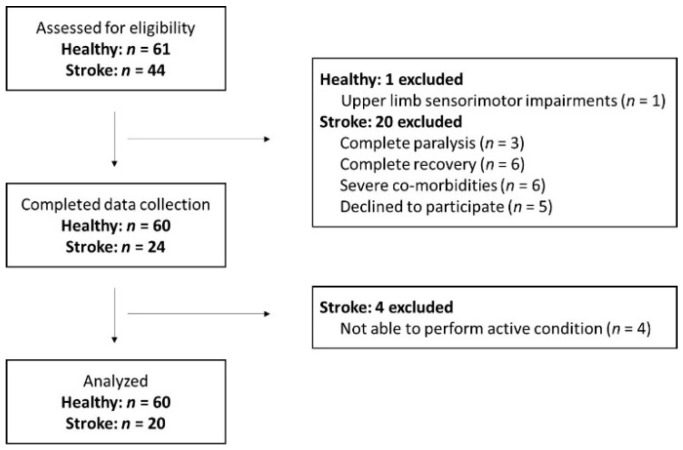
Flowchart of participant inclusion.

**Figure 2 brainsci-12-01005-f002:**
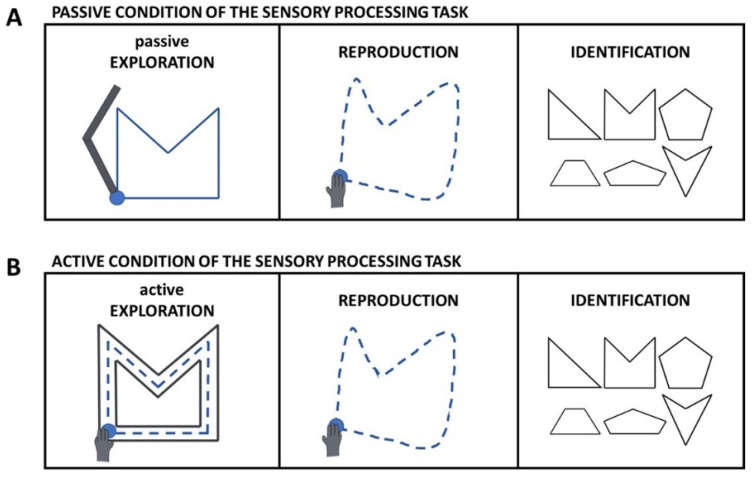
(**A**). Passive condition of the sensory processing task. Left panel: Passive exploration of the shape with the affected or non-dominant arm. Middle panel: Reproduction of the shape with the less affected or dominant arm. Right panel: Identification of the explored shape. (**B**). Active condition of the sensory processing task. Left panel: Active exploration of the shape with the affected or non-dominant arm. Middle panel: Reproduction of the shape with the less affected or dominant arm. Right panel: Identification of the explored shape. Left and middle panels: Blue dashed line = active movement; Solid blue line = passive movement; Solid black line = virtual walls delimiting the shape.

**Figure 3 brainsci-12-01005-f003:**
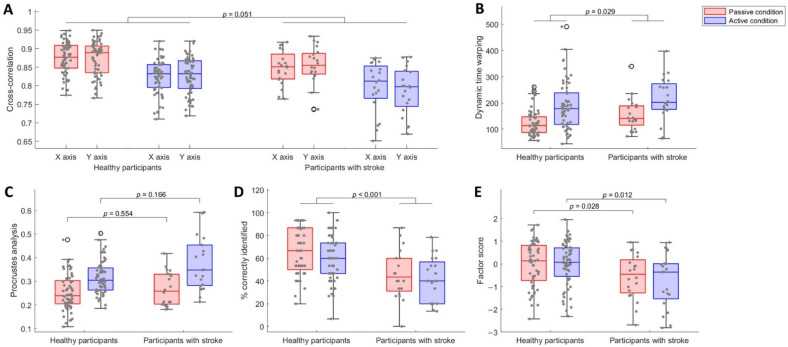
Results of the passive (in red) and active (in blue) sensory processing assessments. (**A**). Main effect for between-group analysis of three-way ANOVA for cross-correlation on X and Y axes. (**B**). Main effect for between-group analysis of two-way ANOVA for dynamic time warping. (**C**). Simple main effects for between-group analysis of two-way ANOVA for Procrustes analysis. (**D**). Main effect for between-group analysis of two-way ANOVA for the percentage of correctly identified shapes. (**E**). Between-group comparison of factor scores using independent *t*-tests.

**Table 1 brainsci-12-01005-t001:** Participant characteristics.

	Healthy Participants (*n* = 60)	Participants with Stroke (*n* = 20)	*p*-Value
Median age in years (IQR)	62.36 (57.30–66.74)	59.47 (48.94–65.69)	0.284
Male, *n* (%)	28 (47)	13 (65)	0.200
Right-handed, *n* (%)	56 (93)	18 (90)	0.637
Ischemic stroke, *n* (%)		11 (55)	
Median time since stroke in years (IQR)		2.65 (1.79–3.36)	
Left hemiparesis, n (%)		13 (65)	
Median MoCA score 0–30 (IQR)	28 (27–29)	27 (26–28)	0.017 *
Median FM-UE score 0–66 (IQR)	66 (64–66)	63 (48–64)	<0.001 *
Median ARAT score 0–57 (IQR)	57 (57–57)	57 (36–57)	<0.001 *
Median BI score 0–20 (IQR)	20 (20–20)	20 (19–20)	<0.001 *
Median EmNSA score 0–40 (IQR)	40 (39–40)	39 (37–40)	0.004 *
Median st-NSA score 0–22 (IQR)	21 (21–22)	21 (20–22)	0.116
Median PTT average in mA (IQR)	2.5 (2.2–3.2)	3.2 (2.1–3.7)	0.218
Median TDT score 0–25 (IQR)	18 (16–18)	15 (11–17)	0.002 *
Median TDT area under the curve (IQR)	64.83 (52.75–75.17)	37.41 (13.62–56.16)	<0.001 *
Median WPST average error in degrees (IQR)	6.23 (5.43–8.51)	9.20 (7.65–11.40)	0.002 *
Median fTORT score 0–42 (IQR)	41 (41–42)	40 (38–41)	0.002 *
Median VGR non-dominant/affected arm task score (IQR)	0.98 (0.73–1.39)	2.46 (1.62–4.03)	<0.001 *
Median VGR dominant/less affected arm task score (IQR)	0.86 (0.60–1.35)	1.45 (0.98–1.84)	0.005 *
Median VGR inter-limb task score (IQR)	0.73 (0.36–1.20)	1.48 (1.18–3.59)	<0.001 *
Median APM non-dominant/affected arm task score (IQR)	0.78 (0.34–1.27)	0.96 (0.52–2.09)	0.075

* *p*-value < 0.050. Abbreviations: *n* = number; MoCA = Montreal cognitive assessment; FM-UE = Fugl-Meyer upper extremity assessment; ARAT = action research arm rest; BI = Barthel index; EmNSA = Erasmus modified Nottingham sensory assessment; PTT = perceptual threshold of touch; st-NSA = stereognosis section of original Nottingham sensory assessment; TDT = tactile discrimination test; WPST = wrist position sense test; fTORT = functional tactile object recognition test; VGR = visually guided reaching; APM = arm position matching.

**Table 2 brainsci-12-01005-t002:** Overview of clinical assessments.

	Modality Tested	Scoring: Range, Cut-Off and Interpretation	Reliability	Validity
Erasmus modified Nottingham sensory assessment [23]	Somatosensory function: exteroception, proprioception and sensory processing	Ordinal score: 0–40 Higher scores = better	Good to excellent [23]	
Stereognosis subscale of original Nottingham sensory assessment [24]	Somatosensory function: sensory processing	Ordinal score: 0–22 Higher scores = better	Moderate to good [25]	
Perceptual threshold of touch [26]	Somatosensory function: exteroception	Smallest detectable stimulus: 0–10 mA Lower scores = better Adult norms available [27]	Good [26]	
Tactile discrimination test [28]	Somatosensory function: sensory processing	Number of correct answers: 0–25 Area under the curve: <60.25 indicates abnormal performance Higher scores = better	Good [28]	Discriminative validity [28]
Wrist position sense test [29]	Somatosensory function: proprioception	Average error: >10.37° indicates abnormal performance Lower scores = better	Good [29]	Discriminative validity [29]
Functional tactile object recognition test [30]	Somatosensory function: sensory processing	Ordinal score: 0–42 Higher scores = better		Discriminative validity [31]
Fugl-Meyer upper extremity assessment [32]	Motor function	Ordinal score: 0–66 Higher scores = better	Excellent [33]	Good convergent validity [33]
Montreal cognitive assessment [34]	Cognitive function	Ordinal score: 0–30, <26 indicates mild cognitive deficits Higher scores = better		Discriminative validity [34]
Star cancellation test [35]	Visuospatial neglect	Number of crossed out stars: 0–54, <44 indicates visuospatial neglect Higher scores = better	Good [36]	Moderate convergent validity [35] Discriminative validity [35,36]
Action research arm test [37]	Motor activity	Ordinal score: 0–57 Higher scores = better	Excellent [37]	Excellent convergent validity [37]
Barthel index [38]	Activities of daily life	Ordinal score: 0–20 Higher scores = better	Good to excellent [39,40]	Good convergent validity [40]

**Table 3 brainsci-12-01005-t003:** Factor loadings of the reproduction and identification parameters.

	Factor Loading	
	Passive Condition	Active Condition
Cross-correlation on X-axis	0.87	0.87
Cross-correlation on Y-axis	0.92	0.87
Dynamic time warping	0.45	0.46
Procrustes analysis	0.92	0.91
% correctly identified	0.68	0.64

**Table 4 brainsci-12-01005-t004:** Number of participants with stroke showing abnormal performance as compared to healthy participants on the passive and active conditions of the sensory processing task.

		Mean (95% CI) of Healthy Participants	Participants with Stroke Outside of 95% CI Indicating Worse Performance, *N* (%)
Passive condition	Cross-correlation on X-axis	0.87 (0.86–0.89)	14 (70)
	Cross-correlation on Y-axis	0.87 (0.86–0.88)	10 (50)
	Dynamic time warping	122.67 (110.62–134.72)	12 (60)
	Procrustes analysis	0.25 (0.23–0.27)	9 (45)
	% correctly identified	66.16 (61.06–71.26)	16 (80)
	Factor score	0.00 (−0.25–0.25)	11 (55)
Active condition	Cross-correlation on X-axis	0.83 (0.81–0.84)	10 (50)
	Cross-correlation on Y-axis	0.83 (0.81–0.84)	13 (65)
	Dynamic time warping	188.50 (165.11–211.89)	9 (45)
	Procrustes analysis	0.32 (0.30–0.33)	11 (55)
	% correctly identified	59.68 (54.58–64.79)	15 (75)
	Factor score	0.00 (−0.25–0.25)	11 (55)

Abbreviations: *n* = number; CI = confidence interval.

**Table 5 brainsci-12-01005-t005:** Correlation coefficients between factor scores of the robot-based sensory processing tasks, and clinical and robot-based assessments of somatosensory function, motor function, cognitive function and activities.

		Factor Score Passive Condition		Factor Score Active Condition	
		r_W_	95% CI	r_W_	95% CI
Somatosensory function	EmNSA-SB total score	0.38	(−0.08 0.70)	0.40	(−0.05 0.72)
	EmNSA total score	0.40	(−0.05 0.72)	0.49	(0.06 0.77)
	st-NSA total score	0.16	(−0.31 0.56)	0.32	(−0.14 0.67)
	PTT average	0.25	(−0.21 0.63)	0.14	(−0.32 0.55)
	TDT total score	0.52	(0.10 0.78)	0.65	(0.29 0.85)
	TDT area under the curve	0.53	(0.11 0.79)	0.71	(0.38 0.87)
	WPST average error	−0.40	(−0.72 0.05)	−0.43	(−0.73 0.02)
	fTORT total score	0.37	(−0.09 0.70)	0.47	(0.03 0.75)
	APM affected arm task score	−0.20	(−0.59 0.27)	−0.37	(−0.70 0.09)
Motor function	FM-UE total score	0.37	(−0.08 0.70)	0.54	(0.13 0.79)
	VGR affected arm task score	−0.44	(−0.74 0.00)	−0.57	(−0.81 −0.17)
	VGR less affected arm task score	−0.03	(−0.47 0.42)	−0.06	(−0.49 0.39)
	VGR inter-limb task score	−0.49	(−0.77 −0.06)	−0.62	(−0.84 −0.25)
Cognitive function	MoCA total score	−0.15	(−0.56 0.31)	−0.31	(−0.66 0.16)
Activities	ARAT total score	0.30	(−0.16 0.66)	0.46	(0.03 0.75)
	BI total score	0.24	(−0.23 0.61)	0.34	(−0.12 0.68)

Black = negligible correlation (r_W_ < 0.30); Red = low correlation (r_W_ = 0.30–0.50); Yellow = moderate correlation (r_W_ = 0.50–0.70); Green = high correlation (r_W_ > 0.70). Note that lower values are associated with better performance for PTT, WPST, APM and VGR. For all other outcomes, higher values are associated with better performance. Abbreviations: r_W_ = Winsorized correlation coefficient; CI = confidence interval; EmNSA-SB = sharp-blunt discrimination subscale of Erasmus modified Nottingham sensory assessment; EmNSA = Erasmus modified Nottingham sensory assessment; st-NSA = stereognosis section of original Nottingham sensory assessment; PTT = perceptual threshold of touch; TDT = tactile discrimination test; WPST = wrist position sense test; fTORT = functional tactile object recognition test; APM = arm position matching; FM-UE = Fugl-Meyer upper extremity assessment; VGR = visually guided reaching; MoCA = Montreal cognitive assessment; ARAT = action research arm rest; BI = Barthel index.

## Data Availability

Data are available upon reasonable request from the author L.S.

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
