# Peer review of "Development and Validation of a Novel Robot-Based Assessment of Upper Limb Sensory Processing in Chronic Stroke"

_brainsci, 2022, doi:10.3390/brainsci12081005_

Round 1

Reviewer 1 Report

Thanks for the opportunity to evaluate this manuscript. I find the authors presented a very clear and well documented manuscript. I have only one criticism.

Discussion:

The authors indicate, as a limitation,  that mild to moderately motor involved participants with stroke are included in this study. Based on ARAT and FM-UE scores, I would argue that only mildly or very mildly motor involved participants were included. This severely limits the generalizability of these results, but more importantly the usefulness of this measure in this population. This should be addressed.

Reviewer 2 Report

The current manuscript introduces a novel assessment of sensory assessmnt for chronic stroke patients. 

Sensory deficits is common in upper limb after stroke. Moreover, sensory recovery could be conditional for motor recovery [Zandvliet SB, Kwakkel G, Nijland RHM, van Wegen EEH, Meskers CGM. Is Recovery of Somatosensory Impairment Conditional for Upper-Limb Motor Recovery Early After Stroke? Neurorehabilitation and Neural Repair. 2020;34(5):403-416.].

The scale is robotic-based evaluate exteroception, proprioception and sensory processing. 

Author Response

We would like to thank Reviewer 2 for their time and kind comments.

We indeed include the key reference mentioned by this reviewer (reference 10).

Reviewer 3 Report

Are the inclusion criteria for stroke survivors related to thresholds in the assessment scales?

Did the participants have any anti-gravity support while performing the assessments?

LL. 117-118 ‘Participants were allowed to explore each shape once, at a self-determined speed within a time limit of 30 seconds. Each line of the explored shapes was between 2.92 cm and 14.14 cm in length’. Can you please clarify to which phase this exploration is related?

In par 2.4. Please provide a brief description of the other robot-based assessment.
For example:
- L. 130 4-target visually guided reaching test was performed bilaterally. Did they perform this task with both arms sequentially or simultaneously?
- L. 133
9-target arm position matching test was performed. How this test was performed? One arm was used as a reference and the other for the matching or the same for both phases sequentially?
What about practice and visual feedback?
Which are the parameters that you computed for the assessments? What are they evaluating(e.g., time, speed, accuracy, precision)?
Which were the instructions that you gave to the participants?

L.145 ‘hand velocity threshold of 0.02 m/s’ how did this threshold have been chosen? Why not a customized threshold according to the max velocity?

Fig.2 Increasing the font of the labels will improve the readability of the figure

Par. 3.5 Add reference and a clear definition of how you define the strength of the correlation (e.g., low-moderate or high-very high). The strength is referred to as the correlation usually, not the correlation coefficients. Can you please clarify this point?

A robotic-based assessment of the sensory ability of healthy participants and stroke survivors has been made in Ballardini et al. 2018

Discuss more the results on the correlation to the other robot-based assessments. 

Reviewer 4 Report

Comments for “Development and validation of a novel robot-based assessment 2 of upper limb sensory processing in chronic stroke”

The manuscript takes up an interesting research, and the study is mostly complete. But, a few minor modifications are needed.

The minor review is as follows.

In the Abstract:

We strongly encourage authors to use the subheadings given. Background and Objectives: Place the question addressed in a broad context and highlight the purpose of the study. Materials and Methods: Describe briefly the main methods or treatments applied, including the study population description. Results: Summarize the article's main findings. Conclusions: Indicate the main conclusions or interpretations. The abstract should be an objective representation of the article, it must not contain results which are not presented and substantiated in the main text and should not exaggerate the main conclusions.

(Notes: Subheadings should be Italics)

1. Introduction

In the introduction, please add the hypothesis of the study at the end.

2. Materials and Methods

Add a note about sample size.

Insert your flowchart as a picture.

Discussion

Authors didnot discuss the limitation of their study in the discussion section which can strengthen the manuscript.
